# Gamification as an approach to improve resilience and reduce attrition in mobile mental health interventions: A randomized controlled trial

**Silja Litvin**[1]*, **Rob Saunders**[2], **Markus A. Maier**[1], **Stefan Lüttke**[3]

**1** Department of General Psychology II (Emotion and Motivation), Ludwig Maximilian University of Munich, Munich, Germany, **2** Research Department of Clinical, Educational and Health Psychology, Centre for Outcomes Research and Effectiveness, University College London, London, United Kingdom, **3** Department of Clinical Psychology and Psychotherapy, University of Tübingen, Tübingen, Germany

* silja@psycapp.com

**Data Availability Statement:** We have published our data with OSF under the link: https://osf.io/v6g3s/.

## Abstract

Forty percent of all general-practitioner appointments are related to mental illness, although less than 35% of individuals have access to therapy and psychological care, indicating a pressing need for accessible and affordable therapy tools. The ubiquity of smartphones offers a delivery platform for such tools. Previous research suggests that gamification—turning intervention content into a game format—could increase engagement with prevention and early-stage mobile interventions. This study aimed to explore the effects of a gamified mobile mental health intervention on improvements in resilience, in comparison with active and inactive control conditions. Differences between conditions on changes in personal growth, anxiety and psychological wellbeing, as well as differences in attrition rates, were also assessed. The *eQuoo* app was developed and published on all leading mobile platforms. The app educates users about psychological concepts including emotional bids, generalization, and reciprocity through psychoeducation, storytelling, and gamification. In total, 358 participants completed in a 5-week, 3-armed (*eQuoo*, "treatment as usual" cognitive behavioral therapy journal app, no-intervention waitlist) randomized controlled trial. Relevant scales were administered to all participants on days 1, 17, and 35. Repeated-measures ANOVA revealed statistically significant increases in resilience in the test group compared with both control groups over 5 weeks. The app also significantly increased personal growth, positive relations with others, and anxiety. With 90% adherence, *eQuoo* retained 21% more participants than the control or waitlist groups. Intervention delivered via *eQuoo* significantly raised mental well-being and decreased self-reported anxiety while enhancing adherence in comparison with the control conditions. Mobile apps using gamification can be a valuable and effective platform for well-being and mental health interventions and may enhance motivation and reduce attrition. Future research should measure *eQuoo*'s effect on anxiety with a more sensitive tool and examine the impact of *eQuoo* on a clinical population.

**Funding:** there has been no financial funding or sources of financial support externally or internally for this work that could have influenced its outcome.

**Competing interests:** I have read the journal's policy and the authors of this manuscript have the following competing interests: I, Silja Litvin, am the majority shareholder of the company PsycApps Limited, which developed eQuoo, the game used in the test group for this trial. The corresponding authors have no other conflicts of interest associated with this publication, and there has been no significant financial support for this work that could have influenced its outcome. This does not alter our adherence to PLOS ONE policies on sharing data and materials.

## Introduction

Mental disorders are highly prevalent and are a major contributor to years lived with disability [1]. Indeed, depression ranks as the single greatest contributor to disability worldwide [2]. Beyond individual burden, mental disorders entail considerable economic costs exceeding expenditures for other common diseases, such as cardiovascular diseases [3]. With the increasing trends in psychological distress, suicide, and self-harm among young people [4], effective treatment and prevention are urgently needed. Despite the high demand and need for services, the majority of individuals in need of mental health services receive no treatment [5, 6]. Moreover, although mental health promotion has long been advocated as a means of preventing mental disorders, prevention has been neglected [7, 8]. This is somewhat surprising given the fact that prevention reduces both economic costs [3] and risk of recurrence of symptoms [9]. This discrepancy between evidence-based interventions for serious mental health conditions and their lack of delivery and implementation in the healthcare system has been referred to as the "research-practice gap" [10].

mHealth (mobile health care), the delivery of mental health treatment using smartphones and wearables, presents an opportunity to narrow the research-practice gap by addressing barriers to treatment at the individual level. Indeed, recent research suggests that the most common reasons for not seeking mental health treatment are respondents' desire to manage their problems on their own, stigma associated with seeking psychological help, and perceptions of treatment as inconvenient or ineffective [11–13]. On the other hand, mobile health apps incorporate unique features that have the potential to circumvent the aforementioned barriers. Apps enable users to help themselves in a self-selected and anonymous setting, thereby enhancing empowerment and reducing stigma. In 2018, approximately 2.9 billion people owned a smartphone worldwide [14], making smartphones a promising and convenient platform from which to provide telemedical care, especially to those who have no or little access to conventional treatment. Consequently, the market for health-related mobile applications is rapidly expanding, with more than 318,000 health apps available when last counted [15].

Despite the utility of mobile health apps as a low-threshold mechanism to deliver treatment having been frequently praised [16], evidence for the efficacy of commercially available apps is scarce [10]. The paucity of such data notwithstanding, there is a growing body of randomized controlled trials (RCTs) showing the potential of mobile mental health apps to reduce symptoms of depression, anxiety, and stress [17, 18]. Further, many individuals show interest in technology to deal with health-related conditions. For example, survey data indicate that roughly three-quarters of psychiatric patients expressed an interest in using a smartphone to monitor their mental health and willingness to do so [19, 20]. However, translation from interest to actual use of apps to manage health-related concerns remains a challenge, as patients tend to use apps only rarely in their daily routine [21]. Examining objective app usage data, a systematic review by Baumel et al. revealed that only a small portion of individuals actually use apps targeting depression, anxiety, and emotional well-being over a long time period [22]. Moreover, smartphone-delivered interventions for mental health problems suffer from low engagement, with mean attrition rates ranging from 9.6% for interventions targeting general stress to 50.0% for apps focusing on insomnia relief [23]. High attrition rates have previously been reported for both web-based interventions and app-based interventions [24–27], implicating attrition as a considerable issue in remote healthcare delivery. Given the potential for mobile applications to restore and maintain mental health, it is vital that the research community investigate means of improving engagement and adherence.

Gamification (i.e., the application of game design elements and mechanisms in non-game contexts) [28–30] is an emerging field in the context of e-Health. There are a number of game

elements, and the most commonly used features in mental well-being apps are *levels/progress feedback*, *points/scoring/ranks*, *rewards/prizes* and *narrative/theme* [30]. With respect to the aim of this study, gamification has been acknowledged as a promising strategy to increase engagement with an intervention and thereby to reduce attrition and augment the intended treatment effects [28, 30, 31]. Although the specific psychological and neural mechanisms require further research, it is assumed that gamification enhances intrinsic motivation by satisfying basic psychological needs of autonomy, competence, and relatedness [30, 31]. Indeed, playing video games is related to fulfilment of these needs [32]. However, game elements, such as point accumulation are linked to socially mediated reinforcement, (i.e., obtaining a goal involves motivation via extrinsic factors). There is limited research on the impact of gamification on attrition in mobile health apps. In an RCT with type 2 diabetes patients, Höchsmann et al. [33] found significant increases in motivation and adherence to physical exercise in a gamified health app intervention group compared with a control group (one-time lifestyle counseling). Interestingly, the intervention group reported significantly higher enjoyment and perceived competence, which lends support to the assumption that gamification influences behavior change via intrinsic motivation. In contrast, a systematic review by Brown et al. [34] could not find a positive effect of gamification on program adherence to web-based mental health intervention. It is important to note, however, that in contrast to mobile mental health apps, most web-based interventions in the literature incorporated only one game element and that adherence has shown a positive correlation with number of game elements. Further, research on the effect of gamification in well-being interventions is scarce. A recent trial showed that playing a videogame may reduce treatment-resistant depressive symptoms in patients receiving antidepressant medication [35]. However, participants were not randomized in that trial, and the sample involved only a subgroup of those with major depression. To our knowledge, there is only one study directly testing the impact of gamified versus non-gamified versions of the same web-based-intervention on well-being [36]. Results of that study showed no difference between interventions on positive affect. However, participants in the gamified intervention group did exhibit elevated involvement, flow, interest, and inspiration.

Another factor to consider in the development of a mental health app is its purpose. Fostering mental health includes not only treatment but also prevention (universal, selective, and indicated) and maintenance interventions (relapse prevention and after-care) [7]. Although mental health, defined as hedonic and eudaimonic well-being [37], is not the opposite of mental illness, the two are interrelated [38]. This becomes more obvious when looking at the various features that constitute well-being (e.g., positive affect, personal growth, positive relations with others, and social integration) [38]. By contrast, negative affect and dysfunctional social relations have been identified as factors contributing to the development of mental disorder and to the persistence of symptoms. For instance, trait negative affect predicted symptoms of depression, social anxiety, panic, and worry in a momentary assessment study [39]. Depression has also frequently been linked to maladaptive social functioning [40, 41], and the risk of depression is significantly negatively correlated with social support and relationship quality [42]. On the other hand, a recent systematic review indicated that social support and a large, diverse social network serve a protective role against depression [43]. Resilience is an individual's ability to adapt and recover following adverse events [44] and is another psychological construct besides well-being worth considering in the context of mental health apps. Resilience may reduce susceptibility to depression and improve stress regulation [45]. Moreover, resilience has been linked to superior mental health and lower self-reported stress [46]. In summary, it is reasonable to assume that promoting well-being and resilience is relevant in the prevention of psychological burden and clinical symptoms.

## Objectives and hypotheses

The purpose of this study was to examine the impact of gamification in a mobile mental health well-being app on self-reported resilience among those using mobile mental health services. Furthermore, we aimed to explore the effect of gamification on measures of well-being (positive relations with others, personal growth, and anxiety as a marker of negative affect) as well as attrition, in the context of a mobile mental health app. To achieve this goal, we conducted a randomized controlled trial in which participants used a gamified mobile mental health app, used a non-gamified mobile mental health app, or were assigned to the waitlist group.

The app *eQuoo* was designed to utilize the influence of gamification on psychoeducation, the internalization of learned concepts and their implementation in real-life settings, which is a large part of the therapeutic process [27]. The leap from learning psychological concepts within a game to learning the skills needed to enhance the user's well-being within a psychological intervention is not that broad [47]; thus, we hypothesized that users completing all elements would also benefit significantly from gamified interventions. In *eQuoo*, 8 gamification elements were implemented, as defined by Tondello and colleagues [48],: (1) *Levels*: one level a week where the player has to learn two skills and implement them correctly in a choose your own adventure story in order to unlock the following level; (2) *progress feedback* in the form of a "progress island map"; (3) *points* as coins; (4) varying *narratives* for each level; (5) *personalization* as generalized feedback of their Big Five personality type; (6) *customization* in allowing users to choose their own mini-avatar; (7) *mini games* where players can deepen skillsets, and finally (8) *badges* ranging from "beginner" to "self-aware". These are the same gamification tools used in video games and, increasingly, in mobile health technology [30]. These tools have been shown to enhance exposure to digital therapies, which has a direct link to effectiveness [49]. Thus, we hypothesized that participants in the *eQuoo* group would exhibit significantly increased self-reported resilience in using the mobile mental health game relative to participants in the cognitive behavioral therapy (CBT) app group. We further hypothesized significantly increased personal growth, positive relations with others, as well as decreased anxiety and attrition, in those receiving the eQuoo intervention compared to the control conditions.

## Materials and methods

### Sample and setting

Participants in this study were identified from among the population of Bosch UK employees and were recruited as part of their well-being benefit program between June 1 and July 30, 2019. To estimate the required sample size and *a priori* power calculation was conducted using G*Power software for a 3 (between-subjects) × 3 (within-subjects) design with an *a priori* effect size estimate of f = 0.1 (small) for the primary outcome (resilience, see below) and an achieved power of 0.80. The calculation indicated that an overall sample size of at least 327 participants would be needed to detect a time x group interaction effect with 80% probability if one was present. Considering the high rate of attrition reported in internet-based interventions [25] which is on average between 43–99%, emails were sent out to 2,500 participants with an invitation to participate in the study. This recruitment procedure was designed to best ensure that at least 327 participants would complete the 5-week trial, taking potentially low uptake and high attrition into consideration. In addition to the email, posters containing a QR code leading to the study's landing page were hung in the cafeterias of 20 Bosch UK locations.

This RCT was approved by the Ethics Committee of Ludwig Maximilian University of Munich, Germany, and is registered as trial DRKS00016039. The study was conducted according to the principles expressed in the Declaration of Helsinki.

## Eligibility criteria

For the participants to be as representative of the general population as possible, the only eligibility criteria were (a) access to a smartphone or tablet device, (b) ability to read English, and (c) an iTunes or Google Play account. Information about previous mental health difficulties and previous treatment or use of well-being apps was not obtained. Only participants who completed the final assessment were included in the presented analyses.

## Data collection

Data were collected using LimeSurvey, which is a highly certified and secure open source online scientific data collection software. All data collection processes were reviewed and approved by the Ethics Committee of Ludwig Maximilian University of Munich, Germany.

## Measures

**Well-being.** To capture a range of constructs that have been related to well-being, we used a range of validated measures to capture resilience, personal growth, interpersonal relationship skills and anxiety.

*Resilience Research Centre—Adult Resilience Measure*. The Resilience Research Centre—Adult Resilience Measure (RRC-ARM), Section C is a 12-item scale developed by Ungar in 2002 [50]. The test offers ratings on a 5-point Likert-type scale *(1 = Not at All, 2 = A Little, 3 = Somewhat, 4 = Quite a Bit, 5 = A Lot)*, and is a screening tool designed to measure the resources (individual, relational, communal, and cultural) available to individuals that may bolster their resilience. Resilience has been identified as one of the major factors in maintaining mental well-being and dealing with stressors in a healthy way, as well as reducing risk-taking behaviors [51]. Because 4 questionnaires were to be administered and questionnaire fatigue was to be avoided, the short version was chosen, rather than the longer 28-item version. From that, the 5-item Likert scale was chosen, rather than the 3-item Likert scale. Cronbach's alpha reliability = 0.953 [52]. The ARM was considered the primary outcome in this study.

*Positive relations with others subscale*. Ryff's Scales of Psychological Well-Being (PWB) was developed in 1989 [53], and is a 6×14-item scale of psychological well-being (84 items total) designed to measure the dimensions of autonomy, environmental mastery, personal growth, positive relations with others, purpose in life, and self-acceptance. The test offers ratings on a 6-point Likert-type scale with positively and negatively scored items *(1 = strongly disagree, 2 = disagree somewhat, 3 = disagree slightly, 4 = agree slightly, 5 = agree somewhat, 6 = strongly agree)*. The subsection "Positive Relations with Others" was selected to avoid questionnaire fatigue while still addressing the isolation and feelings of personal disconnect that have been reported to be major factors in mental unhappiness [54] in the population targeted in this trial. Additional factors include inability to connect and build meaningful relationships, which raises doubts about the ability of an individual to make meaningful contributions to their community. Cronbach's alpha reliability = 0.88 [55].

*Personal growth initiative scale*. The Personal Growth Initiative Scale (PGIS) was designed by Robitschek in 1999 [56]. It is a 9-item Likert-type scale with 6 possible answers to each question *(1 = Definitely disagree, 2 = Mostly disagree, 3 = Somewhat disagree, 4 = Somewhat agree, 5 = Mostly agree, 6 = Definitely agree)*. One element of being a well-adjusted, functioning adult able to contribute to society is personal growth. Positive personal development allows people to achieve goals they have set, such as finishing college, maintaining employment, and advancing in their career. Cronbach's alpha reliability = 0.89 [56, 57]

*One-item anxiety scale*. The one-item anxiety scale measures the current anxiety level by asking how anxious someone feels at the moment [58]. The scale exhibits a high correlation

with the State-Trait Anxiety Inventory [59], which is a commonly used measure to assess state and trait anxiety [60]. The scale can be administered as a visual analog scale with higher scores indicating greater anxiety or a 5-point Likert scale (1 = not at all anxious, 2 = a little anxious, 3 = moderately anxious, 4 = very anxious, 5 = extremely anxious). In this study we used the 5-point Likert scale.

**Attrition.** Attrition was defined as not completing the assessments at both day 17 (t2) and day 35 (t3) of the trial. This is in line with previous research according to which attrition occurs when a participant fails to complete the study protocol associated with the intervention [24, 61].

## Test group intervention

The test group was asked to download a free app called *eQuoo*, available on all major app platforms. The app has 5 levels and is intended to be used over a 5-week period. For each level, the player learns two psychological skills extracted from CBT therapies, positive psychology therapies, and systemic therapies. Each skill is taught in a tutorial format by an avatar named "Dr. Joy", who introduces the player to the game, and explains the processes and skills involved. Each skill is introduced in 3 steps: 1) Dr. Joy explains a skill while cartoon stick figures help visually represent the concepts, 2) possible reactions to the skill are laid out and explained by Dr. Joy using the stick figures, and 3) the player's knowledge of the skill is tested by two characters, Jasmine and Noah, who remain the same throughout the tutorials. Tutorials progress across a series of real-life scenarios such as applying for a job or feeling insecure about a relationship. Once the player masters two skills, a choose-your-own-adventure story opens up where the player is confronted with helping the story's characters complete a challenge. The stories are presented in different genres, including a fantasy story, a sci-fi story, an office story, a love story, and a family holiday story. Each story has two types of questions with three answers to choose from. The first type of question is the "concept question", which tests an individual's mastery of the skills learned in the tutorial, and the second is a personality question based on the Big Five Personality Test [62]. The concept question can be answered incorrectly twice before the story ends in a humorous disaster with feedback as to why the choices resulted in failure of that level. The OCEAN questions lead to feedback about the player's personality. The skills taught in *eQuoo* are as follows: emotional bids [63], generalization [64], action bias [65], confirmation bias [65], catastrophizing [66], halo effect [67], reciprocity [68], expectancy effect [69], courtesy bias [70], and self-serving bias [71]. Details of each skill and instance of personality feedback can be read in a library feature within the app.

## Control group intervention

Different CBT techniques and positive psychology methods have been linked to mental well-being [72], including the constructs tested in this paper: resilience [73], interpersonal relationships skills—even in settings as dire as domestic abuse [74], personal growth and various types of anxiety [75]. The control group was asked to download a free app called *CBT Thought Diary*, which was developed by MoodTools, is an evidence-based tool and is available on all major app platforms. As there are currently no evidence-based mental health games targeting mental well-being, *CBT Thought Diary*, based on CBT and Positive Psychology, was chosen as "treatment as usual", being representative of the majority of existing mobile mental health apps. In the app, the client can track their mood by choosing one of 5 smiley icons, label their emotions, identify negative and distorted thinking patterns, perform a typical CBT exercise, and maintain a mood diary and gratitude journal [76]. Because the *eQuoo* group would be

spending an average of 10–15 min in the game each week, the control group was asked to use the *CBT Thought Diary* for 10 min per week.

### Waitlist group

The waitlist group received no intervention but completed the questionnaires at the same time points as the control and test groups. After completion of the trial, they were debriefed on the trial results and were given a link to both *eQuoo* and the *CBT Thought Diary* app.

### Design

A mixed factorial 3 (condition) ×3 (time) repeated measures design was applied. Participants were randomly assigned to a condition (test group vs. control vs. waitlist group). Across the study period, measures were conducted at the beginning (t1), and again at 17 (t2) and 35 (t3) days after the trial began.

### Procedure

Two weeks before starting data collection, we sent a recruitment email to 2500 Bosch UK employees using their work email addresses provided by the company's HR department. Three days later, a reminder email was sent. The email contained a link to sign up for the trial that led to a landing page where the consent form was presented. Below the form was a button stating, "I understand and accept". When the button was clicked, participants were randomly assigned the test, control, or waitlist control group. Random assignment was achieved using a randomization generator provided by random.org. The randomness comes from atmospheric noise, which for many purposes is better than the pseudo-random number algorithms typically used in computer programs [77]. Moreover, the first enrollment form asked participants for their demographic data, as well as an email address so that they could be contacted at the beginning of the study. After a 2-week sign-up period, the study was launched, and 3 different emails were sent to each group asking them to download the app and fill out the 4 questionnaires. The test group was specifically asked to play only one level per week.

Each week, the test group was sent an email requesting them to play a single level of *eQuoo*, and the control group was asked to use the *CBT Thought Diary* for 10 min. This resulted in the test and control group receiving a total of 5 emails over the 5-week trial period. The waitlist group only received an email with the questionnaires alongside the other two groups: at the beginning (t1), on day 17 (t2), and on day 35 (t3) of the trial.

### Statistical analyses

Statistical analysis was performed with SPSS 25.0 software. Characteristics of the three groups were compared using chi-square independence tests for categorical variables and one-way analysis of variance (ANOVA) models.

To evaluate the impact of the intervention conditions on changes in the primary outcome (resilience–ARM), as well as secondary outcomes (PRWO, PGIS, and Anxiety), 3 (intervention) × 3 (time) repeated-measures ANOVA models were conducted. Bonferroni-corrected pairwise comparisons were conducted to explore the differences between interventions. Paired sample t-tests (two-tailed) were conducted using pre-post (t1 vs. t3), as well as mid-point and post (t2 vs t3) data within each intervention to explore the impact of the interventions on change in the measures of interest and effect sizes (Cohen's d) were calculated. The odds of attrition were calculated for each intervention, and odds ratios (as well as the p-value and 95%

confidence intervals) were estimated by comparing the odds between interventions. A p-value of less than 0.05 was considered statistically significant.

# Results

## Participants

Details of enrollment organized according to the CONSORT guidelines are shown in Fig 1. Descriptive statistics of participants who completed the final questionnaire administration are presented in Tables 1 and 2. Most participants were 35–44 years of age, male, and white.

## Effects of treatment

**Primary outcome.** To test the effect of the intervention on resilience, 3 (intervention groups) × 3 (time points) repeated-measures ANOVA was performed. Bonferroni-corrected post hoc tests were conducted to explore the differences between the interventions.

The repeated-measures ANOVA of the primary outcome (ARM score) yielded a significant main effect of intervention ($F_{(2,350)} = 8.51$, $p < 0.001$, $\eta p2 = 0.046$) and a significant interaction between intervention and time ($F_{(4,698)} = 3.34$, $p = 0.01$, $\eta p2 = 0.019$), but not a main effect of time ($F_{(2,350)} = 2.66$, $p = 0.07$, $\eta p2 = 0.015$). Mean ARM scores over time for each intervention group are presented in Fig 2. *eQuoo* participants exhibited greater increases in scores compared with the CBT journal and waitlist control groups. Bonferroni-corrected post hoc comparisons indicated a significant difference in means between the *eQuoo* and waitlist group ($p < 0.001$), but not between *eQuoo* and the CBT journal group, or the waitlist control and CBT journal group. Within intervention pre (t1)–post (t3) effect sizes were calculated, indicating a small effect for the *eQuoo* group (M(SD)pre = 49.32(5.60), and M(SD) post = 50.87(5.31), $d_{rm} = -0.37$), compared with a limited effect for the CBT journal (M(SD) pre = 48.56(6.68), and M(SD) post = 48.82(6.70), $d_{rm} = -0.06$) and waitlist groups (M(SD) pre = 47.35(7.36), and M(SD) post = 47.08(7.51), $d_{rm} = 0.06$). Further Bonferroni-corrected post hoc tests indicated significant differences in scores between the *eQuoo* group and the waitlist group at t1 ($p = 0.049$), t2 ($p < 0.001$), and t3 ($p < 0.001$), whereas the difference between the CBT journal and *eQuoo* (or the waitlist group) was not statistically significant at any time point ($p > 0.05$).

*Secondary outcomes.* Repeated-measures ANOVAs were also conducted for each of the three secondary outcome measures (personal growth, interpersonal relationship skills and anxiety). For each outcome 3 (intervention groups) × 3 (time points) repeated-measures ANOVA was performed., with bonferroni-corrected post hoc tests were conducted to explore the differences between the interventions.

ANOVA of Ryff's Scales of Psychological Well-Being (RPRS) score yielded a significant main effect of intervention ($F_{(2,342)} = 3.26$, $p = 0.04$, $\eta p2 = 0.019$) and a significant interaction between intervention and time ($F_{(4,682)} = 6.73$, $p < 0.001$, $\eta p2 = 0.038$), but not a main effect of time ($F_{(2,341)} = 2.82$, $p = 0.06$, $\eta p2 = 0.016$). Mean RPRS score over time for each intervention group is presented in Fig 3. Results indicated that *eQuoo* participants displayed greater increases in scores compared to the CBT journal and waitlist control groups. Bonferroni-corrected post hoc comparisons showed that the main effect of intervention came from the significant difference in means between the *eQuoo* and waitlist group ($p = 0.03$), but not between *eQuoo* and the CBT journal group or the waitlist control and CBT journal group. Within intervention pre (t1)–post (t3) effect sizes were calculated, indicating a small effect for the *eQuoo* group (M(SD)pre = 61.67(11.73), and M(SD) post = 64.53(10.74), $d_{rm} = -0.42$), compared to a limited effect for the CBT journal (M(SD)pre = 61.08(12.18), and M(SD) post = 61.15(12.71), $d_{rm} = -0.01$) and waitlist groups (M(SD)pre = 58.88(13.93), and M(SD) post = 58.35(13.111),

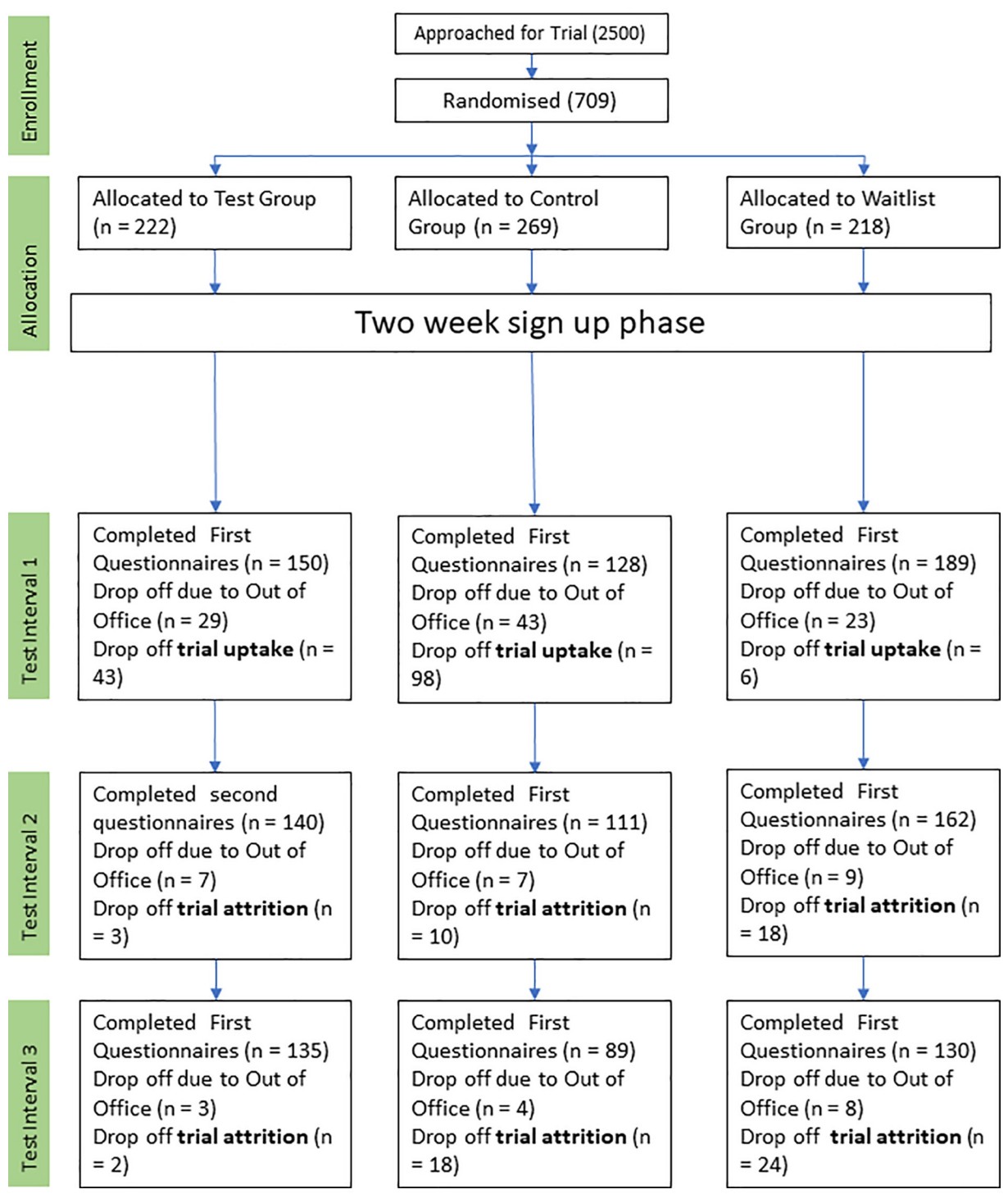

CONSORT (2010) Flow Diagram for a 3-Arm Study

**Fig 1. CONSORT flow diagram for a 3-arm study.**

**Table 1. Outcome measures of the sample.**

| Outcome Measures | | Mean | Sd | Mean | Sd | Mean | Sd | p-value^ |
|---|---|---|---|---|---|---|---|---|
| ARM | Time 1 | 49.32 | 5.60 | 48.56 | 6.68 | 47.35 | 7.36 | 0.051 |
| | Time 3 | 50.66 | 5.53 | 48.79 | 6.16 | 47.13 | 7.45 | <0.001 |
| | Time 5 | 50.87 | 5.31 | 48.82 | 6.70 | 47.08 | 7.51 | <0.001 |
| RPRS | Time 1 | 61.61 | 11.79 | 61.08 | 12.18 | 58.82 | 13.89 | 0.180 |
| | Time 3 | 61.89 | 11.11 | 61.00 | 11.89 | 59.29 | 13.85 | 0.231 |
| | Time 5 | 64.53 | 10.66 | 61.15 | 10.66 | 58.44 | 13.11 | <0.001 |
| PGIS | Time 1 | 38.43 | 8.07 | 37.21 | 8.88 | 36.41 | 7.74 | 0.129 |
| | Time 3 | 39.51 | 6.90 | 37.82 | 8.23 | 37.02 | 8.15 | 0.030 |
| | Time 5 | 42.15 | 6.43 | 39.42 | 7.78 | 37.70 | 7.99 | <0.001 |
| Anxiety | Time 1 | 1.05 | 0.96 | 1.07 | 1.15 | 1.32 | 1.15 | 0.078 |
| | Time 3 | 0.90 | 0.96 | 0.99 | 1.00 | 1.25 | 1.00 | 0.012 |
| | Time 5 | 0.88 | 0.98 | 0.88 | 0.85 | 1.21 | 1.07 | 0.011 |

ARM, Adult Resilience Measure; Anxiety, a One-Item Likert Scale; PGIS, Personal Growth Initiative Scale; RPRS, Ryff's Scales of Psychological Well-Being; SD, standard deviation.

*From chi-square tests of independence.

^From one-way ANOVAs.

$d_{rm}$ = 0.09). Further Bonferroni-corrected post hoc tests indicated significant differences in scores between the *eQuoo* group and the waitlist group at t3 only (p < 0.001).

ANOVA of the PGIS score revealed a significant main effect of intervention ($F_{(2,348)}$ = 5.81, p = 0.003, ηp2 = 0.03) and time ($F_{(2,347)}$ = 40.32, p < 0.001, ηp2 = 0.19) and a significant interaction between intervention and time ($F_{(4,694)}$ = 3.99, p = 0.003, ηp2 = 0.02). Mean PGIS score over time for each intervention group is presented in Fig 4. These data suggest that *eQuoo* participants had greater increases in scores compared with the CBT journal and waitlist control groups, and Bonferroni-corrected post hoc comparisons showed that the main effect of intervention came from the significant difference in means between the *eQuoo* and waitlist group (p = 0.002), but not between *eQuoo* and the CBT journal group or the waitlist control and CBT journal group. Within-intervention pre (t1)–post (t3) effect sizes were calculated,

**Table 2. Characteristics of the sample.**

| Demographics | | eQuoo(n = 135) | | CBT Journal(n-95) | | Waitlist(n = 130) | | |
|---|---|---|---|---|---|---|---|---|
| | | n | % | n | % | n | % | p-value* |
| Age | 16–24 | 20 | 14.8% | 11 | 12.4% | 14 | 10.8% | 0.43 |
| | 25–34 | 27 | 20.0% | 21 | 23.6% | 41 | 31.5% | |
| | 35–44 | 46 | 34.1% | 22 | 24.7% | 35 | 26.9% | |
| | 45–54 | 33 | 24.4% | 28 | 31.5% | 29 | 22.3% | |
| | 55–64 | 9 | 6.7% | 7 | 7.9% | 10 | 7.7% | |
| | 64+ | 0 | 0.0% | 0 | 0.0% | 1 | 0.8% | |
| Gender | Female | 60 | 44.4% | 26 | 29.2% | 54 | 41.5% | 0.119 |
| | Male | 74 | 54.8% | 63 | 70.8% | 76 | 58.5% | |
| | Missing | 1 | 0.7% | 0 | 0.0% | 0 | 0.0% | |
| Ethnicity | White | 119 | 88.2% | 78 | 87.6% | 113 | 86.9% | 0.769 |
| | Asian | 7 | 5.2% | 5 | 5.6% | 8 | 6.2% | |
| | Black | 2 | 1.5% | 1 | 1.1% | 0 | 0.0% | |
| | Other | 7 | 5.2% | 5 | 5.6% | 9 | 6.9% | |

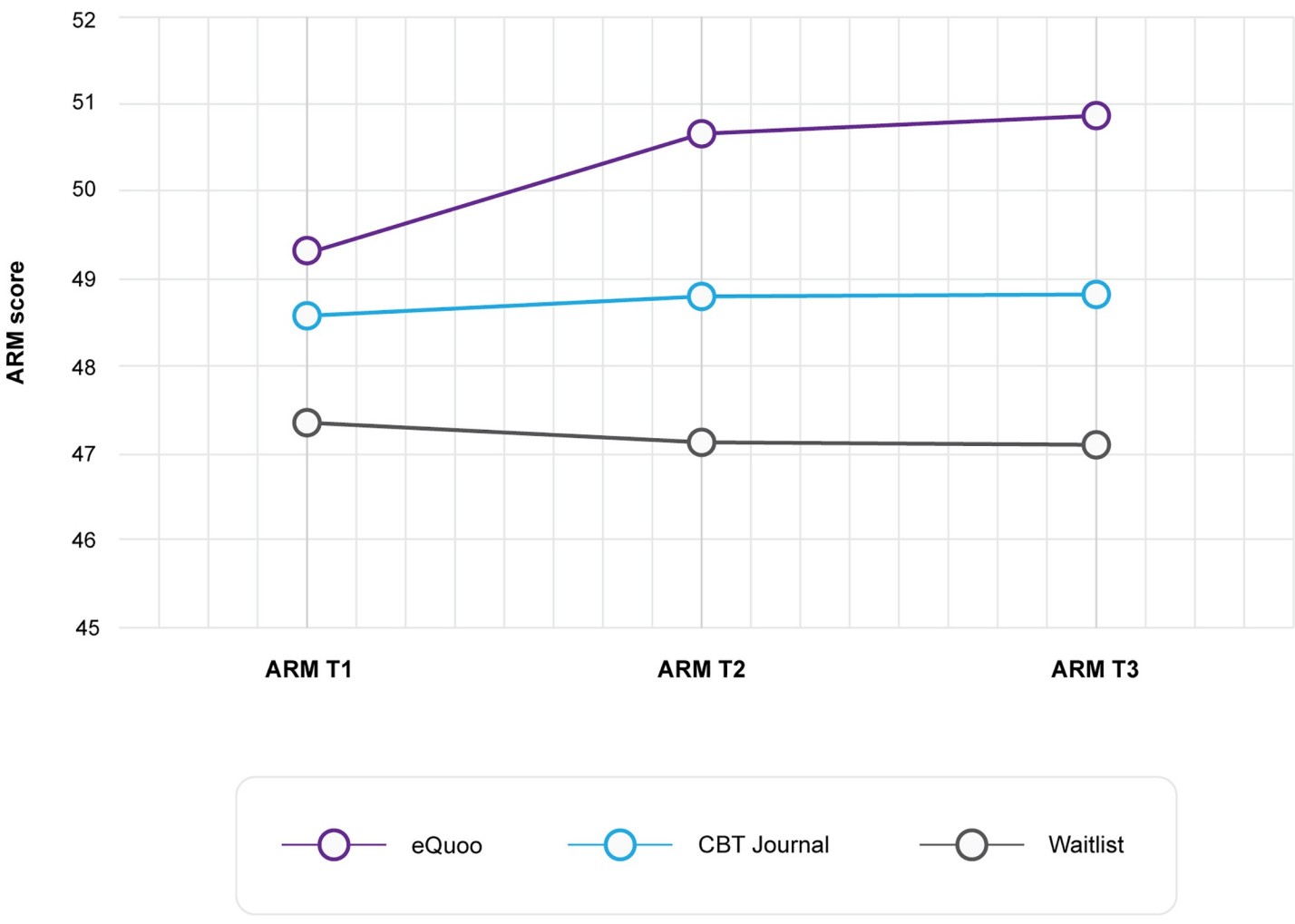

**Fig 2. ARM results at each time period.**

indicating a medium effect for the *eQuoo* group (M(SD)pre = 38.42(8.07), and M(SD) post = 42.15(6.43), $d_{rm}$ = -0.67), compared with a small effect for both the CBT journal group (M(SD)pre = 37.21(8.88), and M(SD) post = 39.42(7.78), $d_{rm}$ = -0.48) and the waitlist group (M(SD)pre = 36.36(7.75), and M(SD) post = 37.7(7.99), $d_{rm}$ = -0.25).

Further Bonferroni-corrected post hoc tests indicated significant differences in scores between the *eQuoo* group and the waitlist group at t2 (p = 0.02) and t3 (p < 0.001), and between the CBT journal and *eQuoo* at t3 (p = 0.032).

ANOVA of the single-item anxiety score revealed a significant main effect of intervention ($F_{(2,342)}$ = 4.972, p = 0.007, $\eta p2$ = 0.03) and time ($F_{(2,341)}$ = 4.74, p = 0.009, $\eta p2$ = 0.03), but not an interaction between intervention and time ($F_{(4,682)}$ = 0.44, p = 0.78, $\eta p2$ = 0.003). Fig 5 shows the change in mean anxiety score across groups, indicating that anxiety was highest for the waitlist group, and the change was uniform across groups. The within-intervention pre (t1)–post (t3) effect sizes were small or limited for each group (*eQuoo* group: (M(SD) pre = 1.05(0.97), M(SD) post = 0.88(0.98), $d_{rm}$ = 0.20); CBT journal: (M(SD)pre = 1.07(1.04), M(SD) post = 0.88(0.85), $d_{rm}$ = 0.19); waitlist control: (M(SD)pre = 1.33(1.14), M(SD) post = 1.21(1.08), $d_{rm}$ = 0.12)).

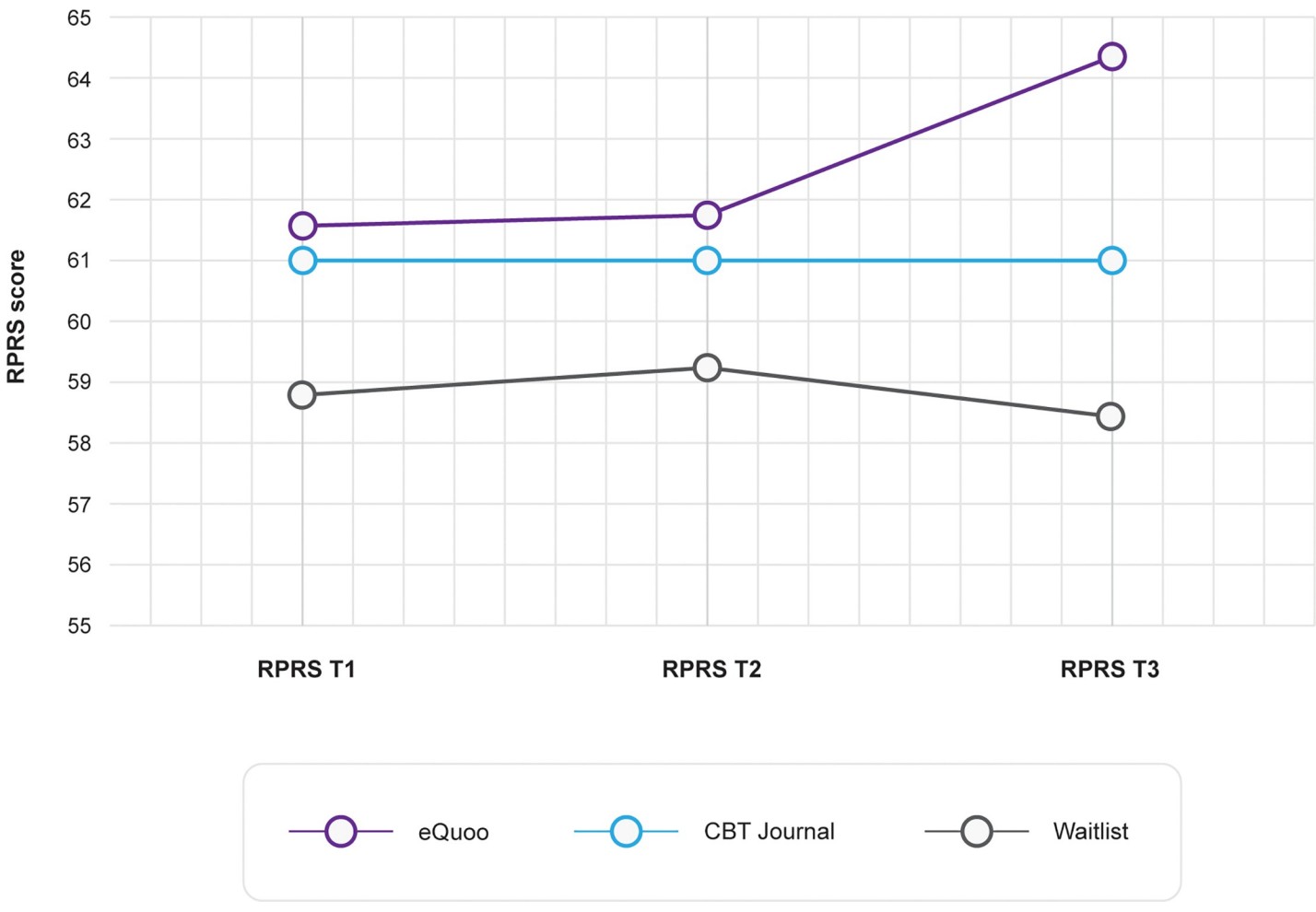

**Fig 3. RPRS scores at each time point.**

Further Bonferroni-corrected post hoc tests indicated significant differences in scores between the *eQuoo* group and the waitlist group at t1 (p = 0.009) and t2 (p = 0.037), and differences between the CBT journal and waitlist at t3 (p = 0.038).

## Attrition

It is important to note that 225 potential participants were lost due to their use of their business email addresses for initial sign up, resulting in their inability to receive the trial reminder emails with the questionnaires during their annual leave. These attrition numbers have been excluded from the trial attrition numbers, as the attrition was imposed by external factors. The attrition results for participants in the test group, control group, and waitlist group are shown in Fig 6.

Of the 222 participants in the test group, the questionnaire was answered by 150 participants at t1, 140 at t2, and 135 at t3, corresponding to attrition of 6.7% and 10%, respectively.

Of the 269 participants in the control group, the questionnaire was answered by 128 participants at t1, 111 at t2, and 89 at t3, corresponding to attrition of 13.3% and 30.5%, respectively.

Of the 218 participants in the waitlist group, the questionnaire was answered by 89 participants at t1, 162 at t2, and 130 at t3, corresponding to attrition of 14.3% and 31.2%, respectively.

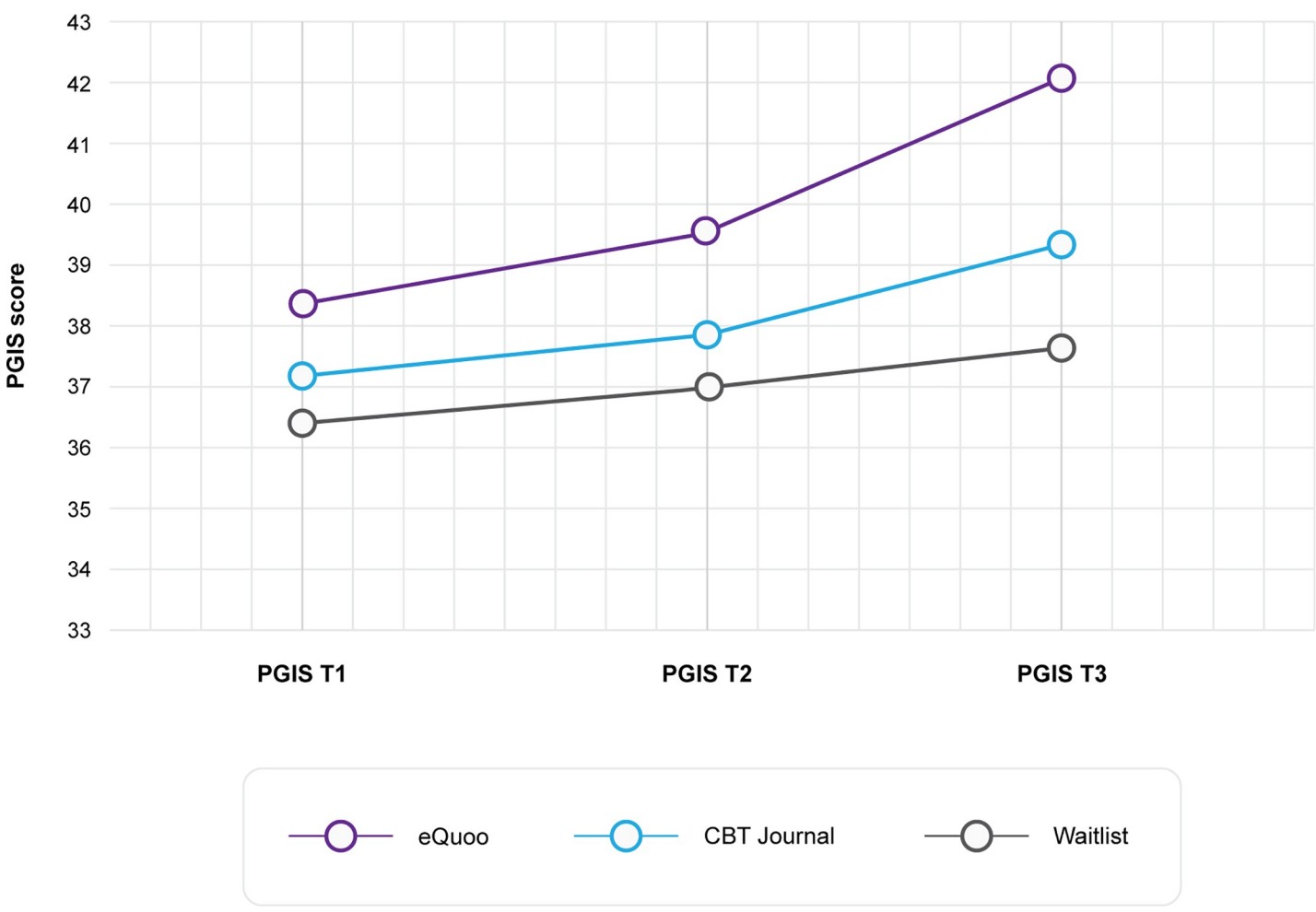

**Fig 4. PGIS scores at each time point.**

The odds of remaining in the study were calculated for each intervention group. Overall, the odds of remaining in the study at t3 for those who completed the t1 questionnaire were 3.94 times higher in the test group compared with the control group (odds ratio = 3.944, 95% confidence interval [CI] = 2.053–7.576, p < 0.001; 10% vs. 30.5% attrition) and 4.08 times higher in the test group compared with the waitlist group (odds ratio = 4.084, 95% CI = 2.207 to 7.561, p < 0.001; 10% vs. 31.2% attrition).

Because the trial was conducted during the summer, when many Bosch UK associates took their annual leave, 225 "out of office" emails were received in response to the emails sent during the 3 time periods. The owners of the accounts sending those messages had no access to their accounts while out of the office and were not evaluated. This resulted in a 31% external attrition rate of participants who otherwise might have completed the trial.

## Discussion

This randomized controlled study investigated the impact of a mental well-being mobile app, *eQuoo*, which incorporates gamification in order to (1) reduce attrition in mobile health services and (2) improve mental health well-being—defined as resilience, personal growth, interpersonal skills, and current anxiety level. Consistent with our hypothesis, results indicated a

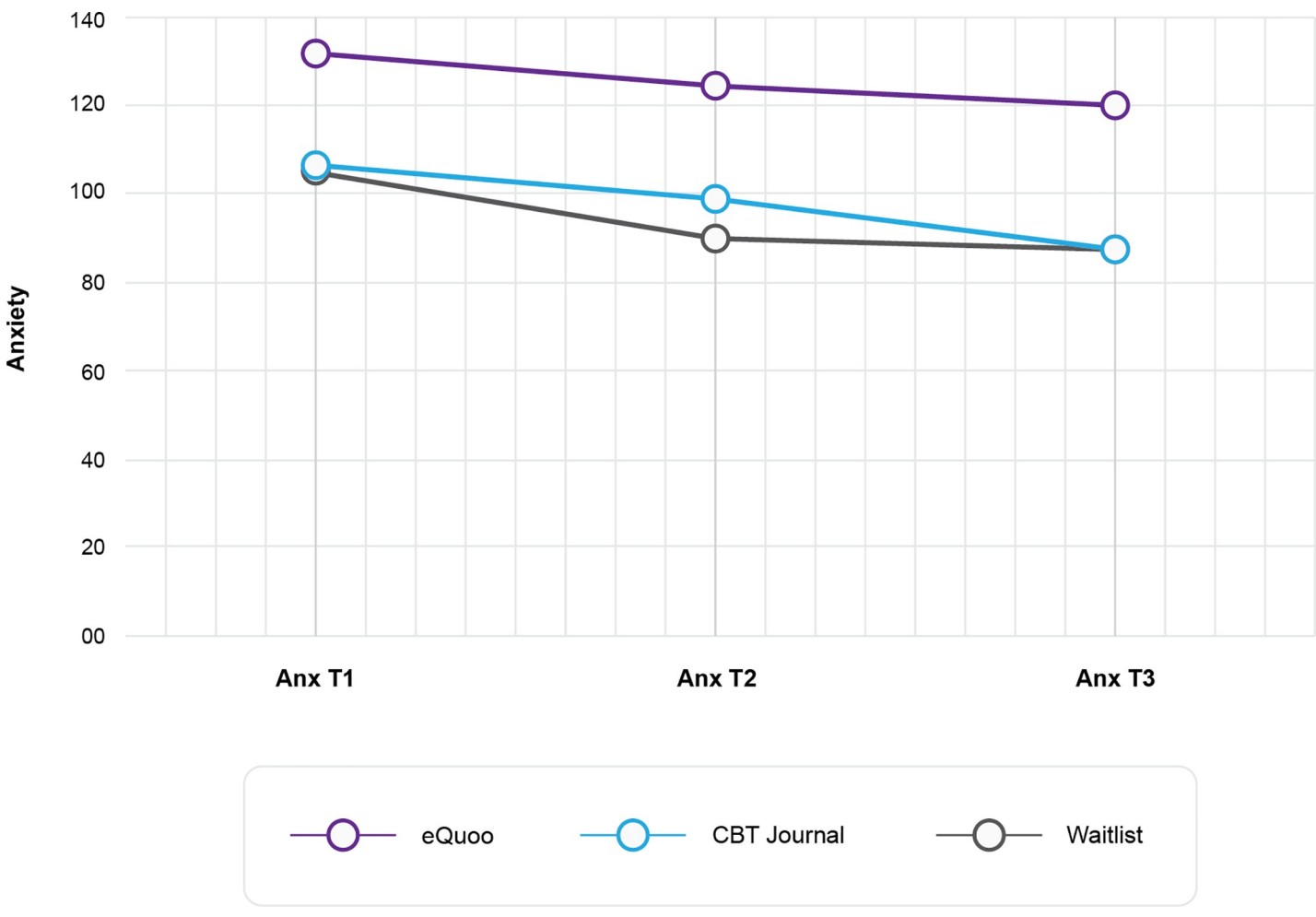

**Fig 5. Anxiety scores at each time point.**

significantly lower attrition rate in the *eQuoo* app group compared with both the control intervention group (CBT journal app) and the no-intervention waitlist group. The odds of study protocol adherence was approximately 4 times higher in the *eQuoo* app group, suggesting that using a gamified mental well-being app is associated with a higher adherence to the study protocol than using a non-gamified mental health app or receiving no active treatment.

We also found partial evidence for the efficacy of gamification on well-being. Participants in the *eQuoo* app group exhibited higher scores on positive relations with others in comparison with participants in either the CBT journal app or waitlist group. However, this effect was driven by the difference between the *eQuoo* app group and the waitlist group. Participants using the *eQuoo* app did not significantly differ in relationships with others compared with those using the CBT journal app. Moreover, substantial gains in the *eQuoo* group occurred in the final two weeks of app usage, but not during the initial weeks of the study. Personal growth scores were higher in the *eQuoo* app group compared with both control groups. Importantly, personal growth differed significantly between participants in the *eQuoo* app versus waitlist control group at t2 and t3, whereas the difference in personal growth between *eQuoo* app and CBT journal app groups was significant at t3 only. Beyond a main effect of intervention, we also found a main effect of time on personal growth. Specifically, personal growth increased

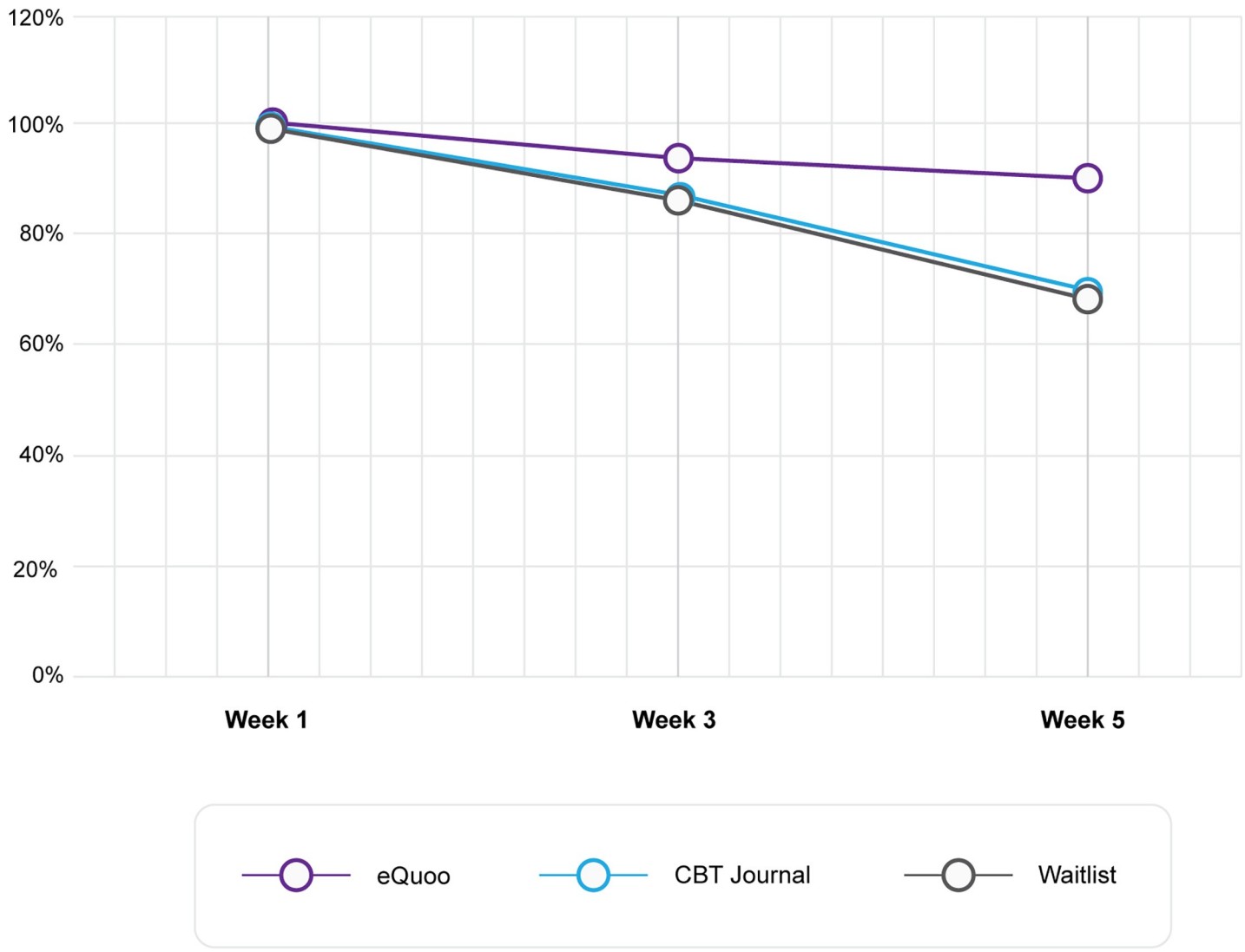

**Fig 6. Percentage of participants who submitted questionnaire responses at each time point.**

from t1 to t3 in all groups. Anxiety showed a uniform decrease across time. In line with our hypothesis, participants using the gamified *eQuoo* app showed a significant decline in anxiety compared with the waitlist group. However, there was no significant difference between the two active intervention groups. Finally, participants in the *eQuoo* group exhibited gains in resilience over the course of the study, but similar to other well-being measures, this effect was only significant for the *eQuoo* app versus waitlist group. In contrast to other well-being measures, resilience measures in the *eQuoo* app group showed significant improvement between t1 and t2, and subsequently remained stable. In summary, participants in the *eQuoo* app group exhibited improvements on all well-being measures during the course of the study. However, apart from a significantly higher personal growth score in the *eQuoo* app group compared with the CBT journal app group at t3, participants using the *eQuoo* app did not perform significantly better in terms of well-being than participants who used a non-gamified mental health app. Thus, we cannot rule out the possibility that the effects were based on not only the use of gamification, but also exposure to the underlying intervention.

Attrition is a major problem in online interventions, and gamification has been suggested as a promising strategy to foster adherence. In a meta-analysis investigating attrition and adherence in smartphone-based mental health interventions, Linardon and Fuller-Tyszkiewicz [61] reported attrition rates of approximately 33% and 42% for short-term and long-term follow-up interventions, respectively. Their results resemble the attrition we found for the CBT journal app group in our study. Importantly, the attrition rate in the group using a gamified mental well-being app was significantly reduced, supporting the idea that gamification may indeed strengthen adherence when using mental health apps. This is in line with a study by Kelders et al. [36], who investigated the impact of a gamified versus a non-gamified web-based well-being intervention. According to the authors, gamification increased involvement and flow when using the intervention. Although we did not test this directly, based on the results by Kelders et al. [36], it is also reasonable to suggest that higher involvement and flow led to a reduced attrition rate in the gamified app group in our study. Importantly, however, that Kelders et al. found that gamification did not increase behavioral engagement in terms of time spent with the intervention or exercises completed [36]. Moreover, there are other potential explanations for the significantly lower attrition rate in the gamified app group, such as the purpose of the app itself. For instance, approximately 83% of participants did not complete the study protocol in a recent app-based mood diary study with a follow-up period of 1-month (similar to the present study) [78]. However, the app in this study was limited to tracking individuals' emotional states and the context in which a specific emotion occurred, whereas both apps used in the present study encompassed more psychoeducational elements. Other research indicates that factors such as personal guidance in internet-delivered interventions [79], socio-demographic factors [80], confidence in the effectiveness of the online intervention [80], usability of the service [25], and perception of the intervention as personally relevant [81] play an important role in understanding attrition in eHealth. In addition, reasons for discontinuation of an intervention vary depending on the stage within the process. Usability is pivotal when considering whether to use a service at all, whereas positive attitudes toward technology, and experiencing a positive impact of the intervention are crucial to long-term adherence [82]. In conclusion, we cannot directly attribute our results to the impact of gamification. Future research should investigate a gamified versus non-gamified version of the same app intervention and control for the aforementioned factors to rule out alternative explanations.

## Gamification and well-being

To date, only a few studies have explored the impact of mobile apps that incorporate gamification to promote well-being. For instance, Lin et al. found a substantial reduction of smoking after use of a CBT-based mobile app employing game elements aimed to boost participants hedonic well-being, inspiration, and empowerment [83]. Although the authors suggest that gamification increased hedonic well-being, in fact, this conclusion cannot be drawn based on the study design with no control condition. Other research has pointed out that playing mobile games can also positively affect individuals' psychological well-being [84]. As in our study, Ahtinen et al. reported positive effects of a gamified mobile app targeting working-aged people to reduce stress and enhance satisfaction with life in a 1-month field study [85]. However, in contrast to our study, neither of the aforementioned studies was an RCT, leaving open the question of whether game elements or other factors caused the gains in well-being. In addition, these were pilot studies with small sample sizes of 6 participants [84] and 15 participants [85], respectively.

Although studies applying gamification in mobile apps for fostering well-being are scarce, there is an emerging body of literature on the use of gamification to treat mental disorders

such as depression and anxiety [86]. The primary aim of gamification is to improve adherence to and motivation in the intervention [86]. For example, a mobile application incorporating ratings and feedback on progress as game elements significantly reduced depressive symptoms [87]. It is important to note that, compared with the majority of other studies, the interventions in this research were personalized based on context features derived from the interactions of each participant with the smartphone, such as location (home vs. work) or level of current physical activity [87]. Thus, the effects revealed in this study may arise from meaningful recommendations of interventions according to personal context rather than gamification of the intervention. To test the effect that gamification would have on mental health, it is necessary to directly compare a gamified versus a non-gamified version of the same intervention. Recent studies implementing such a design have found promising results in favor of gamification. For instance, adding common game elements such as a reward system to a standard web-based intervention to reduce alcohol consumption significantly increased the efficacy of the treatment over a 2-week course [88]. In another study, Kelders et al. [36] found evidence for significantly higher personal involvement and flow in participants who used a gamified versus non-gamified web-based intervention designed to improve well-being in the general population. Moreover, the gamified interventions enhanced positive affect and intrinsic motivation, though these effects were marginally significant [36]. Interestingly, this effect occurred only in a real-life experiment, not in lab conditions, thus highlighting the importance of considering the context in which a gamified treatment is to be used.

## Strengths and limitations

The present study had several strengths. To our knowledge, this is the first follow-up RCT investigating the impact of gamification on improving resilience, mental well-being and adherence in the context of a smartphone app. Notably, compared with prior research, we were able to investigate the effects in a large sample of approximately 470 participants at the beginning of the study. Given that evidence on the effectiveness of mobile apps on well-being in real-life settings is scarce, this study contributes to the literature on smartphone applications for mental health.

Some limitations of this study need to be acknowledged. One is that the mobile mental health apps used in the study had different purposes, making it difficult to directly evaluate the impact of gamification on attrition and well-being. The focus of the app *eQuoo* is to strengthen resilience by implementing gamification. A virtual character accompanies the user as he or she learns skills associated with well-being. In contrast, the app *CBT Thought Journal* aims to reveal the association between dysfunctional thinking and mood, a typical CBT exercise, and it can be used to document initial distress, emotions, and thoughts with respect to a given situation. Subsequently, the user identifies cognitive distortions within their thoughts and writes an alternative interpretation of the situation. Thus, we cannot exclude possibility that the effects we found in this study were attributable not solely to gamification but also to the different purposes of the apps. Moreover, since we did not record the degree to which participants were engaged with the app (e.g. time of usage) it is not possible to evaluate the impact of gamification on nonusage attrition. Although dropout and nonusage attrition may be substantially correlated [24], future studies should seek to consider both attrition outcome measures to increase internal validity. Another limitation is that the external validity of the study is limited for the following reasons. First, since the sample comprised employees of BOSCH UK, the results should be generalized with caution although we found no indication that the sample differs from the general population. Also, although the primary outcome for the study was resilience, the secondary well-being outcomes of the present study, namely, personal growth,

relationships with others, and anxiety, are somewhat restricted compared with the rather broad construct of well-being, and further studies exploring the impact of dropouts would be crucial to the understanding of mHealth interventions. In this study, only completers were included in the measurements, possibly leading to a neglect of data significant to the development of effective mobile interventions.

The platform limitations of a mobile game also contribute to the limitations, as treatment choices were dictated by the usage of gamification on a developer platform called UNITY as much as research on effective interventions. Lastly, there has been a deviation from the original protocol: In the time from first submission of the study protocol on 5.12.2017 until the acceptance on 13.11.2018 the leading author, Silja Litvin, continued research in the field and came to the conclusion that exchanging the wellbeing questionnaire from the Happiness Scale, to The Resilience Research Centre—Adult Resilience Measure (RRC-ARM), would benefit the study's applicable value to society. The changes were appraised not to warrant a re-submission to the ethics committee, as the psychological dimensions remain the same. The study was shortened to 5 weeks from 6 weeks as the app which was being developed at the time of the ethics submission, only made it to 5 levels instead of the planned six due to funding issues.

## Implications

This study provides the first evidence that gamification has the potential to enhance the effects of mobile health and well-being interventions. However, further research should examine the mechanisms by which gamification impacts mental health and well-being in the real world. To do so, an experimental study should be conducted in a field-study setting. Specifically, to attribute causal effects to gamification, it is necessary to utilize a gamified versus a non-gamified version of the same intervention in an RCT. Another direction of future research is to compare different game elements to determine whether specific gamification strategies are associated with more benefits than others. This is in line with Cheng et al. [89], who advocated dissecting gamification features in order to shine light into the "black box". Therefore, it is vital to position future research within theoretical frameworks such as self-determination theory and hypothesized mechanisms for how game elements influence outcomes of interest [89]. We suggest a multimethod approach for well-being, that is, use of various validated outcome measures such as the 5-item World Health Organization Well-Being Index [90], the Positive and Negative Affect Schedule [91] and the Psychological Distress Scale [92]. Moreover, to establish a meaningful relationship between gamification and mental well-being we suggest monitoring well-being using an ambulatory assessment approach in a longitudinal design [93]. For instance, participants could be prompted to answer short surveys on their well-being immediately after they used the respective intervention. Also, it would be possible to ask participants after an event if they applied strategies taught in the intervention. For attrition, we propose incorporating adherence measures that are directly linked to use of the intervention, such as number of logins, usage duration, and completion of modules or levels.

## Conclusion

This study's objective was to investigate the potential of a gamified mobile app to improve psychological resilience compared with a non-gamified mental health intervention app and a waitlist group. We found superiority of the gamified versus non-gamified app in increasing self-reported resilience. The gamified app also increased other measures of wellbeing, as well as reducing anxiety and attrition rates. Future research should compare a gamified versus non-gamified version of the same mobile app as well as incorporate a multimethod approach and ambulatory assessment to study well-being and attrition in real-world settings.

## Supporting information

**S1 Checklist. CONSORT 2010 checklist of information to include when reporting a randomised trial***.
(DOC)

**S1 Protocol.**
(DOCX)

**S2 Protocol.**
(DOCX)

## Author Contributions

**Conceptualization:** Silja Litvin, Markus A. Maier.

**Data curation:** Silja Litvin.

**Formal analysis:** Silja Litvin, Rob Saunders, Markus A. Maier.

**Investigation:** Silja Litvin.

**Methodology:** Silja Litvin, Rob Saunders.

**Project administration:** Silja Litvin, Stefan Lüttke.

**Resources:** Silja Litvin, Markus A. Maier.

**Software:** Silja Litvin.

**Supervision:** Silja Litvin.

**Validation:** Rob Saunders, Stefan Lüttke.

**Visualization:** Rob Saunders.

**Writing – original draft:** Silja Litvin, Stefan Lüttke.

**Writing – review & editing:** Markus A. Maier.

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
