## [Decision Letter · Decision Letter 0]

1 May 2020

PONE-D-19-35191

Gamification as an approach to improve mental well-being and reduce attrition in mobile mental health interventions: A randomized controlled trial

PLOS ONE

Dear Ms Litvin,

Thank you for submitting your manuscript to PLOS ONE. After careful consideration, we feel that it has merit but does not fully meet PLOS ONE’s publication criteria as it currently stands. Therefore, we invite you to submit a revised version of the manuscript that addresses the points raised during the review process.

We would appreciate receiving your revised manuscript by Jun 15 2020 11:59PM. To enhance the reproducibility of your results, we recommend that if applicable you deposit your laboratory protocols in protocols.io, where a protocol can be assigned its own identifier (DOI) such that it can be cited independently in the future. For instructions see: http://journals.plos.org/plosone/s/submission-guidelines#loc-laboratory-protocols

We look forward to receiving your revised manuscript.

Kind regards,

Yutaka J. Matsuoka, MD, PhD

Academic Editor

PLOS ONE

Journal Requirements:

3. Thank you for stating the following financial disclosure: "no"

"I have read the journal's policy and the authors of this manuscript have the following

competing interests: I, Silja Litvin, am the majority shareholder of the company

PsycApps Limited, which developed eQuoo, the game used in the test group for this

trial. The corresponding authors have no other conflicts of interest associated with this

publication, and there has been no significant financial support for this work that could

have influenced its outcome."

5. Please upload a copy of Figure 6, to which you refer in your text on line 436. If the figure is no longer to be included as part of the submission please remove all reference to it within the text.

6. Please include your tables as part of your main manuscript and remove the individual files. Please note that supplementary tables (should remain/ be uploaded) as separate "supporting information" files

Reviewers' comments:

Reviewer's Responses to Questions

**Comments to the Author**

1. Is the manuscript technically sound, and do the data support the conclusions?

Reviewer #1: Partly

Reviewer #2: Partly

Reviewer #3: Partly

2. Has the statistical analysis been performed appropriately and rigorously? 

Reviewer #1: No

Reviewer #2: No

Reviewer #3: No

3. Have the authors made all data underlying the findings in their manuscript fully available?

Reviewer #1: No

Reviewer #2: No

Reviewer #3: No

4. Is the manuscript presented in an intelligible fashion and written in standard English?

Reviewer #1: Yes

Reviewer #2: Yes

Reviewer #3: Yes

5. Review Comments to the Author

Reviewer #1: The present study aimed to explore the effects of a gamified mobile mental health intervention on adherence and psychological measures in comparison with active and inactive control conditions. The topic itself is interesting. However, there are some issues to be handled in the current manuscript.

Major issues:

#1 Which variable was the primary outcome? The attrition rate seems to be the primary outcome. If so, the title should be change to include only the primary outcome.

#2 With regard to the definition of the attrition rate, why was the frequency or time of the usage of the app not used?

#3 Which statistical method was used to test the effect on the attrition rate? The method should be described in the Statistical Analysese section. In addition, the power analysis should be conducted about the attrition rate.

#4 There were so many secondary outcomes. Therefore, the type I error coule be inflated.

Minor issue:

#5 Tables 1a and 1b was interchanged with each other.

Reviewer #2: The current described randomized controlled three-arm trial to determine the efficacy of “eQuoo” to explore the effects of a gamified mobile mental health intervention on adherence, resilience, personal growth, and psychological well-being in comparison to control groups. The trial include both active (CBT) and inactive control group (wait-list). Each subject contributes up to five time points, as a result, repeated measure mixed effect model should be used for both design and analysis of the study. Any efficacy trial must have sample size power justification based on hypothesized effect size, post-hoc analysis or justification is not an acceptable standard. Given the longitudinal nature of the design the study should have been powered to detect the time*group interaction term prospectively with an hypothesized effect size as well as range of ICC values. The most serious drawback of the study is that no sample size and power is provided. It is unclear how many subjects in each group completes the full study duration from the provided CONSORT diagram. As a result, the study cannot qualify as an efficacy trial.

I have few minor points, though they cannot rescue the trial anyway from the deficiencies mentioned above.

1. There is a large dropout in all three arms. Albeit the dropout from active control group is quite higher than other two groups. This needs explanation and possible there could be informative dropout. This will complicate the analysis provided.

2. The analysis should be done using RMANOVA and the time*group interaction term. I am not sure which term is used to claim significance.

3. Since there are three groups it is not clear how multiplicity of hypothesis testing is adjusted for. The rationale of three-arms and it’s resulting implication of efficacy testing is not accounted for.

4. It is not clear how missing data is handled in the analysis and if principle of ITT is followed.

Also not clear how data is availble for sharing as no informtion is provided about where the data resides and how one can acess it.

Reviewer #3: Thank you for giving me the opportunity to review the manuscript entitled “Gamification as an approach to improve mental well-being and reduce attrition in mobile mental health interventions: A randomized controlled trial”. This study examined the efficacy of gamified psychological mobile application in comparison with other cognitive behavioral applications and waiting lists related to improving psychological well-being, resilience, among others. Studies regarding the examination of mobile applications that specifically target mental well-being are limited; hence, this study provides important findings in the field. However, I think several clarifications are needed to assess this study accurately.

I was unable to find any rationale for the choice of treatment strategy selected (combination of CBT, positive psychology, and systemic therapy) to improve the outcomes for specific participants in this study. I understand that gamification is important for retaining the participants; however, I think important aspects for preparing intervention materials are the evidences and theories of the choice of intervention to be gamified.

I searched the record of clinical trial registry for “DRKS00016039,” but was unable to find any record on the internet. I would request the authors to provide a specific link to the CTR record.

The authors stated that they expected an effect size of 0.1 for calculating the sample size. However, what is the primary outcome for estimating the effect size? Without specifying the primary outcome and its time point, it is impossible to calculate the sample size. In addition, please provide the rationale for an effect size of 0.1, with appropriate citations of the previous findings. Please clarify the specific information on the effect size (e.g., within vs. between, comparison with which intervention?)

Although I am not a native English speaker, I think the expression of the following sentence is awkwardly phrased.

Page 9, lines 192-193:“Data were collected using LimeSurvey, which is open source online scientific data collection software that has that is highly certified and secure.”

Please provide a detailed explanation of the deviation from the study protocol. It is standard research conduct (at least in my country) that the researcher modifies the study protocol and receives approval for the modification from the IRB before proceeding with the clinical trial. I fail to understand why the authors did not follow this protocol.

Although the authors found a few differences in some outcomes between the groups, I was unable to understand the clinical importance of these differences observed in this study. For example, how much is the clinical significance (not statistically) of two-point differences in the ARM and PGIS scores?

Please provide the between effect sizes at time-2 and time-3. In addition, 95% confidence interval should be provided to assess the statistical significance of within and between effect sizes.

In my opinion, Figure 2 is not necessary, because it is unusual to show the attrition in a figure. Information provided in the main text is sufficient.

6. PLOS authors have the option to publish the peer review history of their article (what does this mean?). If published, this will include your full peer review and any attached files.

Reviewer #1: No

Reviewer #2: No

Reviewer #3: Yes: Masaya Ito

---

## [Author Response · Author response to Decision Letter 0]

23 Jun 2020

Reviewer #1: 

The present study aimed to explore the effects of a gamified mobile mental health intervention on adherence and psychological measures in comparison with active and inactive control conditions. The topic itself is interesting. However, there are some issues to be handled in the current manuscript.

Major issues:

#1 Which variable was the primary outcome? The attrition rate seems to be the primary outcome. If so, the title should be change to include only the primary outcome.

Author response: We are grateful for the reviewers pointing out the lack of clarity surrounding the outcomes. The primary outcome of the trial is resilience, which is what the original power calculation was calculated for, and the manuscript has been restructured in a number of places to emphasis the primary outcome. Interpersonal relationship skills, personal growth, anxiety and attrition are considered important secondary outcomes. Considering the high rates of attrition in the literature of app-delivered interventions we considering the findings for this study, where the test intervention was shown to considerably reduce the likelihood of attrition, was an important finding, but as it was not the primary outcome we have taken the comment on board and amended the manuscript as appropriate.

#2 With regard to the definition of the attrition rate, why was the frequency or time of the usage of the app not used?

Author response: We strongly agree that the frequency of the usage of the app would be an interesting measure to estimate how much the participants have experienced the intervention. However, the term attrition has been defined in different ways, for example dropout, premature discontinuation, premature termination, non-usage attrition, non-persistence and non-adherence (for an overview refer to van Ballegooijen et al., 2014). In our study we refer to the term dropout attrition, which has been defined in the groundbreaking article by Eysenbach (2005) as “proportion of users who are lost to follow-up over time” (i.e. participants who do not follow the study protocol). In contrast to attrition, adherence describes the degree to which the individual is engaged with an intervention as determined by logins to the program, time spent on the app, number of app modules complete (Donkin et al., 2011, Linardon & Fuller-Tyszkiewicz, 2019) or nonusuage attrition (Eysenbach, 2005). We clarified the definition of attrition applied in the study in the manuscript’s method section. Since both measures of attrition (dropout and nonusage attrition) measure different aspects of attrition we intend to consider nonusage attrition and adherence respectively in addition to dropout attrition in future studies. We amended this aspect in the discussion. 

van Ballegooijen, W., Cuijpers, P., van Straten, A., Karyotaki, E., Andersson, G., Smit, J. H., & Riper, H. (2014). Adherence to Internet-based and face-to-face cognitive behavioural therapy for depression: a meta-analysis. PloS one, 9(7), e100674. https://doi.org/10.1371/journal.pone.0100674

Donkin L, Christensen H, Naismith SL, Neal B, Hickie IB, Glozier N. A systematic review of the impact of adherence on the effectiveness of e-therapies. J Med Internet Res. 2011;13(3):e52.

Linardon J, Fuller-Tyszkiewicz M. Attrition and adherence in smartphone-delivered interventions for mental health problems: A systematic and meta-analytic review. J Consult Clin Psychol. 2019.

Eysenbach G. The law of attrition. J Med Internet Res. 2005;7(1):e11.

#3 Which statistical method was used to test the effect on the attrition rate? The method should be described in the Statistical Analysis section. In addition, the power analysis should be conducted about the attrition rate.

 Author response: Thank you for highlighting this omission. The manuscript has been updated to describe this analysis. In short, the odds ratio, p-value and 95% Cis were calculated by hand. As this was not the primary outcome, prospective power calculation was not performed and therefore we would not present in the manuscript as if they were, and we appreciate the previous lack of clarity around this outcome.

#4 There were so many secondary outcomes. Therefore, the type I error could be inflated.

Author response: This is a fair point, and we appreciate the potential issue here without specifying a primary outcome for the study. In the absence of a primary outcome, we agree that adjustment would be needed, however the literature would suggest that the presence of a primary outcome negates the need for adjustment across outcomes (see for e.g. https://doi.org/10.1186/1471-2288-2-8 ; https://doi.org/10.1093/ije/dyw320). Within the individual RM-ANOVAs, Bonferroni corrections were applied, and this has been made clearer in the manuscript.

Minor issue:

#5 Tables 1a and 1b was interchanged with each other.

Author response: Thank you pointing this out to us, we have amended.

Reviewer #2:

 #1The current described randomized controlled three-arm trial to determine the efficacy of “eQuoo” to explore the effects of a gamified mobile mental health intervention on adherence, resilience, personal growth, and psychological well-being in comparison to control groups. The trial include both active (CBT) and inactive control group (wait-list). Each subject contributes up to five time points, as a result, repeated measure mixed effect model should be used for both design and analysis of the study. Any efficacy trial must have sample size power justification based on hypothesized effect size, post-hoc analysis or justification is not an acceptable standard. Given the longitudinal nature of the design the study should have been powered to detect the time*group interaction term prospectively with an hypothesized effect size as well as range of ICC values. The most serious drawback of the study is that no sample size and power is provided. It is unclear how many subjects in each group completes the full study duration from the provided CONSORT diagram. As a result, the study cannot qualify as an efficacy trial.

Author response: Thank you for these thoughtful comments. There are a few points raised, please allow us to address them one by one:

• repeated measure mixed effect model should be used for both design and analysis of the study. Author response: A repeated-measures ANOVA was employed, which, owing to the use of complete cases only, would result in practically identical results to a mixed effects model.

• Any efficacy trial must have sample size power justification based on hypothesized effect size, post-hoc analysis or justification is not an acceptable standard. Author response: We apologise for the lack of clarity with regard to the power calculation. Resilience was the primary outcome, and we have restructured the manuscript to make this clearer. The power calculation was calculated for resilience, and we have made this clearer in the “Sample and setting” section where the power calculation is presented.

• Given the longitudinal nature of the design the study should have been powered to detect the time*group interaction term prospectively with an hypothesized effect size as well as range of ICC values.: Author response: Thank you for raising. The study was powered to detect the interaction with a small effect, and this is mentioned in the “Sample and setting” section.

• The most serious drawback of the study is that no sample size and power is provided. Author response: The power calculation is presented in the “Sample and setting” section, and we have clarified as per the suggestions above.

• It is unclear how many subjects in each group completes the full study duration from the provided CONSORT diagram. Author response: We included a CONSORT diagram in the original submission but can see that it isn’t completely clear who remains in the study for analysis. We have clarified the CONSORT diagram for this point, highlighting the final ‘n’ in bold and captioning it “Included in Analysis”.. 

I have few minor points, though they cannot rescue the trial anyway from the deficiencies mentioned above.

#2. There is a large dropout in all three arms. Albeit the dropout from active control group is quite higher than other two groups. This needs explanation and possible there could be informative dropout. This will complicate the analysis provided.

Author response: Thank you for raising this. With a mean attrition rate of 23%, the dropout rate is actually lower than average for a mobile mental health interventions In the introduction section we hypothesize that the on average high attrition rates within mHealth solutions could be explained by them not being attractive to end users. We chose an active Control Group using a classic mental health application as a “Treatment as usual” solution alongside the gamified intervention to test that hypothesis. 

#3. The analysis should be done using RMANOVA and the time*group interaction term. I am not sure which term is used to claim significance.

Author response: Thank you for highlighting this lack of clarity, we have amended the manuscript, so this is clear to the reader.

#4. Since there are three groups it is not clear how multiplicity of hypothesis testing is adjusted for. The rationale of three-arms and it’s resulting implication of efficacy testing is not accounted for.

Author response: Please see our response to Reviewer 1, comment #4 with regard to this, and Bonferroni adjustments were applied to post-hoc comparisons between groups.

#5. It is not clear how missing data is handled in the analysis and if principle of ITT is followed.

Author response: Thank you for raising. We did not include data from anyone who did not complete assessment at all three timepoints, and as such this is a completer analysis. The study was adequately powered for this level of attrition. We consider dropouts to have rescinded their consent to participate in the trial, and do not share that data. This has been added to limitations as well. 

Also not clear how data is availble for sharing as no informtion is provided about where the data resides and how one can acess it.

Author response: Thank you for suggesting this - we have published our data with OSF under the link: https://osf.io/v6g3s/

Reviewer #3: 

Thank you for giving me the opportunity to review the manuscript entitled “Gamification as an approach to improve mental well-being and reduce attrition in mobile mental health interventions: A randomized controlled trial”. This study examined the efficacy of gamified psychological mobile application in comparison with other cognitive behavioral applications and waiting lists related to improving psychological well-being, resilience, among others. Studies regarding the examination of mobile applications that specifically target mental well-being are limited; hence, this study provides important findings in the field. However, I think several clarifications are needed to assess this study accurately.

#1 I was unable to find any rationale for the choice of treatment strategy selected (combination of CBT, positive psychology, and systemic therapy) to improve the outcomes for specific participants in this study. I understand that gamification is important for retaining the participants; however, I think important aspects for preparing intervention materials are the evidences and theories of the choice of intervention to be gamified.

Author response: Thank you for this comment. The choice of treatment strategy was dictated by practicality: we searched for evidence-based treatment features that could easily be implemented into a gamified learning model. Due to platform restrictions of an app built on the UNITY game engine the choices were restricted. As of now we cannot discern which treatment features added which impact to the overall treatment leading to a type of ‘Blackbox’ effect. In the upcoming version of eQuoo, this limitation will be approached using MOST Micro Randomised Controlled Trials with real time data (Klasnja, P., Hekler, E. B. & Shiffman, 2015).We have added this to the limitations sections of the paper. 

Klasnja, P., Hekler, E. B., Shiffman, S., Boruvka, A., Almirall, D., Tewari, A., & Murphy, S. A. (2015). Microrandomized trials: An experimental design for developing just-in-time adaptive interventions. Health Psychology: official journal of the Division of Health Psychology, American Psychological Association, 34S(0), 1220–1228. https://doi.org/10.1037/hea0000305

#2 I searched the record of clinical trial registry for “DRKS00016039,” but was unable to find any record on the internet. I would request the authors to provide a specific link to the CTR record.

Author response: We are grateful for this being highlighted. Please find the link to DRKS00016039 here: https://www.drks.de/drks_web/navigate.do?navigationId=trial.HTML&TRIAL_ID=DRKS00016039

#3 The authors stated that they expected an effect size of 0.1 for calculating the sample size. However, what is the primary outcome for estimating the effect size? Without specifying the primary outcome and its time point, it is impossible to calculate the sample size. In addition, please provide the rationale for an effect size of 0.1, with appropriate citations of the previous findings. Please clarify the specific information on the effect size (e.g., within vs. between, comparison with which intervention?)

Author response: We apologise for the lack of clarity surrounding this. Please see our responses to Reviewer 1 comment #1 surrounding this

.#4 Although I am not a native English speaker, I think the expression of the following sentence is awkwardly phrased.

Page 9, lines 192-193:“Data were collected using LimeSurvey, which is open source online scientific data collection software that has that is highly certified and secure.”

Author response: Thank you for pointing this out. This has been amended to: “Data were collected using LimeSurvey, which is a highly certified and secure open source online scientific data collection software”

#5 Please provide a detailed explanation of the deviation from the study protocol. It is standard research conduct (at least in my country) that the researcher modifies the study protocol and receives approval for the modification from the IRB before proceeding with the clinical trial. I fail to understand why the authors did not follow this protocol.

Author response: We appreciate this comment. In the time from first submission of the study protocol on 5.12.2017 until the acceptance on 13.11.2018 the leading author, Silja Litvin, continued research in the field and came to the conclusion that exchanging the wellbeing questionnaire from the Happiness Scale, to The Resilience Research Centre - Adult Resilience Measure (RRC-ARM), would benefit the study’s applicable value to society. The changes were appraised not to warrant a re-submission to the ethics committee, as the psychological dimensions remain the same. The study was shortened to 5 weeks from 6 weeks as the app which was being developed at the time of the ethics submission, only made it to 5 levels instead of the planned six due to funding issues. PI Professor Markus Maier agreed, leading to the deviation from the protocol. We do, however agree that the deviation is unusual and have added these explanations to the limitations.

#6 Although the authors found a few differences in some outcomes between the groups, I was unable to understand the clinical importance of these differences observed in this study. For example, how much is the clinical significance (not statistically) of two-point differences in the ARM and PGIS scores?

Author response: Thank you for raising. The clinical 'value' has been evidenced but if is it large enough to result in a perceived (subjective) value is a good question some clinicians will be asking. Adding a questionnaire that could help answer that problem would be something worth adding to a follow-up study such as Quality of Life or a general questionnaire on mental problems. 

#7 Please provide the between effect sizes at time-2 and time-3. In addition, 95% confidence interval should be provided to assess the statistical significance of within and between effect sizes.

Author response: We have added these as requested.

#8 In my opinion, Figure 2 is not necessary, because it is unusual to show the attrition in a figure. Information provided in the main text is sufficient.

Author response: We appreciate this feedback. We understand that attrition is usually not included in figures, but in this case, considering the significant levels of attrition reported in mobile mental health we would prefer to keep it as it highlights the potential for gamification to increase engagement in interventions

---

## [Decision Letter · Decision Letter 1]

23 Jul 2020

Gamification as an approach to improve resilience and reduce attrition in mobile mental health interventions: A randomized controlled trial

PONE-D-19-35191R1

Dear Dr. Litvin,

We’re pleased to inform you that your manuscript has been judged scientifically suitable for publication and will be formally accepted for publication once it meets all outstanding technical requirements.

Kind regards,

Yutaka J. Matsuoka, MD, PhD

Academic Editor

PLOS ONE

Additional Editor Comments (optional):

The authors have addressed the comments sufficiently. 

Reviewers' comments:

Reviewer's Responses to Questions

**Comments to the Author**

1. If the authors have adequately addressed your comments raised in a previous round of review and you feel that this manuscript is now acceptable for publication, you may indicate that here to bypass the “Comments to the Author” section, enter your conflict of interest statement in the “Confidential to Editor” section, and submit your "Accept" recommendation.

Reviewer #1: All comments have been addressed

Reviewer #3: All comments have been addressed

2. Is the manuscript technically sound, and do the data support the conclusions?

Reviewer #1: Yes

Reviewer #3: Partly

3. Has the statistical analysis been performed appropriately and rigorously? 

Reviewer #1: Yes

Reviewer #3: Yes

4. Have the authors made all data underlying the findings in their manuscript fully available?

Reviewer #1: Yes

Reviewer #3: Yes

5. Is the manuscript presented in an intelligible fashion and written in standard English?

Reviewer #1: Yes

Reviewer #3: Yes

6. Review Comments to the Author

Reviewer #1: All comments have been addressed adequately. The manuscript is technically sound, and the data support the conclusions.

Reviewer #3: Thank you very much for considering my comments. I think the authors sufficiently and transparently responded to the comments and modified the manuscript.

7. PLOS authors have the option to publish the peer review history of their article (what does this mean?). If published, this will include your full peer review and any attached files.

Reviewer #1: **Yes: **Kazuhiro Yoshiuchi

Reviewer #3: **Yes: **Masaya Ito

---

## [Editor Report · Acceptance letter]

6 Aug 2020

PONE-D-19-35191R1 

Gamification as an approach to improve resilience and reduce attrition in mobile mental health interventions: A randomized controlled trial 

Dear Dr. Litvin:

I'm pleased to inform you that your manuscript has been deemed suitable for publication in PLOS ONE. Congratulations! Your manuscript is now with our production department. 

Kind regards, 

on behalf of

Dr. Yutaka J. Matsuoka 

Academic Editor

PLOS ONE